# Soil Mineral Composition and Salinity Are the Main Factors Regulating the Bacterial Community Associated with the Roots of Coastal Sand Dune Halophytes

**DOI:** 10.3390/biology11050695

**Published:** 2022-04-30

**Authors:** Minh Thiet Vu, Almando Geraldi, Hoang Dang Khoa Do, Arif Luqman, Hoang Danh Nguyen, Faiza Nur Fauzia, Fahmi Ikhlasul Amalludin, Aliffa Yusti Sadila, Nabilla Hapsari Wijaya, Heri Santoso, Yosephine Sri Wulan Manuhara, Le Minh Bui, Sucipto Hariyanto, Anjar Tri Wibowo

**Affiliations:** 1NTT Hi-Tech Institute, Nguyen Tat Thanh University, Ho Chi Minh City 70000, Vietnam; vmthiet@ntt.edu.vn (M.T.V.); dhdkhoa@ntt.edu.vn (H.D.K.D.); danhhoang1804@gmail.com (H.D.N.); 2Departement of Biology, Faculty of Science and Technology, Airlangga University, Surabaya 60115, Indonesia; almando.geraldi@fst.unair.ac.id (A.G.); faiza.nur.fauzia-2018@fst.unair.ac.id (F.N.F.); fahmi.ikhlasul.amalludin-2018@fst.unair.ac.id (F.I.A.); aliffa.yusti.sadila-2018@fst.unair.ac.id (A.Y.S.); nabilla.hapsari.wijaya-2018@fst.unair.ac.id (N.H.W.); yosephine-s-w-m@fst.unair.ac.id (Y.S.W.M.); blminh@ntt.edu.vn (L.M.B.); 3Biotechnology of Tropical Medicinal Plants Research Group, Airlangga University, Surabaya 60115, Indonesia; 4Biology Department, Institut Teknologi Sepuluh Nopember, Surabaya 60111, Indonesia; arif.luqman@its.ac.id; 5Generasi Biologi Indonesia (Genbinesia) Foundation, Gresik 61171, Indonesia; herisantoso@genbinesia.or.id; 6Department of Biotechnology, NTT Hi-tech Institute, Nguyen Tat Thanh University, Ho Chi Minh City 70000, Vietnam

**Keywords:** halophyte, sand dune, root microbiome, salt stress, soil mineral, soil fertility management

## Abstract

**Simple Summary:**

Endophytic microbes that reside in roots are involved in resistance against various environmental stresses including high soil salinity and mineral deficiency. To date, the extent of their role in the plant host adaptation to arid, saline, and low nutrient environments such as coastal sand dune ecosystems remains unclear. Here, we present the first characterization study of the bacterial community associated with the roots of *Spinifex littoreus* and *Calotropis gigantea,* two plant species that grow wild across different areas of Parangkusumo coastal sand dune, Indonesia. We correlated the bacterial composition in the root with various soil properties and found that bacterial communities in the root are responsive to changes in soil mineral composition, especially in soil Calcium (Ca), Titanium (Ti), Cuprum (Cu), and Zinc (Zn) content. Some bacteria are also found to be sensitive to soil salinity levels; among them, *Bacillus idriensis* has previously been reported to have a growth promoting effect on plants. Our findings provided valuable information about the main factors that modulate bacterial communities associated with coastal plants and potential bacterial species that might be involved in plant resistance against stresses. Data from this study can be used as the basis for future studies that assess the biological role of endophytic microbes in plant resistance against environmental pressure.

**Abstract:**

Soil salinity and mineral deficiency are major problems in agriculture. Many studies have reported that plant-associated microbiota, particularly rhizosphere and root microbiota, play a crucial role in tolerance against salinity and mineral deficiency. Nevertheless, there are still many unknown parts of plant–microbe interaction, especially regarding their role in halophyte adaptation to coastal ecosystems. Here, we report the bacterial community associated with the roots of coastal sand dune halophytes *Spinifex littoreus* and *Calotropis gigantea,* and the soil properties that affect their composition. Strong correlations were observed between root bacterial diversity and soil mineral composition, especially with soil Calcium (Ca), Titanium (Ti), Cuprum (Cu), and Zinc (Zn) content. Soil Ti and Zn content showed a positive correlation with bacterial diversity, while soil Ca and Cu had a negative effect on bacterial diversity. A strong correlation was also found between the abundance of several bacterial species with soil salinity and mineral content, suggesting that some bacteria are responsive to changes in soil salinity and mineral content. Some of the identified bacteria, such as *Bacillus idriensis* and *Kibdelosporangium aridum*, are known to have growth-promoting effects on plants. Together, the findings of this work provided valuable information regarding bacterial communities associated with the roots of sand dune halophytes and their interactions with soil properties. Furthermore, we also identified several bacterial species that might be involved in tolerance against stresses. Further work will be focused on isolation and transplantation of these potential microbes, to validate their role in plant tolerance against stresses, not only in their native hosts but also in crops.

## 1. Introduction

Soil salinity and mineral deficiency are major abiotic stresses that affect global agricultural production. In some crops, salt stress and mineral deficiency could reduce average yields by more than 50% [1,2]. Soil salinization also leads to constant reduction of arable lands, with around 800 million hectares of agricultural land currently affected by salt stress [3]. High soil salinity could lead to both ionic and osmotic stress [4]. Ionic stress is mainly caused by excessive intracellular sodium (Na^+^) accumulation that causes a deficiency of essential ions such as potassium (K^+^), affecting protein synthesis and conformation. Ionic stress also induces the production of reactive oxygen species (ROS) that lead to cellular oxidative stress and damage. In addition to ionic stress, salt stress also induces osmotic stress, which leads to reduced water uptake and dehydration [4,5]. 

Most major crops such as wheat, maize, and rice are sensitive to salinity and mineral deficiency. Exposure to salt and mineral deficiency has been known to reduce germination rates, seedling survival, and plant productivity [6,7]. On the other hand, halophytes are salt-tolerant plants that can complete their life cycle in an environment with a salt concentration above 200 mM NaCl [8,9]. The tolerance is mainly attributed to the genetic makeup of the plants, which allows them to produce specific sets of proteins, transporters, and anatomical and morphological modifications that enhance resistance to salinity [9,10,11]. In addition to plant adaptation, it is reported that the root microbiome could also promote resistance to abiotic stresses, including salinity and mineral deficiency. Microbes can be found both on the surfaces (epiphytic) and inside of the roots (endophytic). Together, they constitute the overall microbiome composition in the root [12]. Different species of epiphytic and endophytic bacteria associated with the roots of various halophytic plants are reported to be able to ameliorate the negative effects of salinity and mineral deficiency and promote growth [13,14,15,16,17]. It is suggested that root-associated bacteria could help plants thrive in saline and nutrient-depleted environments by facilitating nitrogen fixation [18], increasing nutrient availability and uptake [17], inducing antioxidant production [19,20], and producing certain metabolites and hormones that have growth-promoting effects on plants [21,22]. 

Interestingly, the beneficial effects of root bacteria are not restricted to their natural hosts. Several publications have reported that bacteria isolated from the roots of halophytes could also enhance tolerance to stresses and promote growth when inoculated into the roots of crops. Sharma et al. (2016) reported that five bacterial isolates from the root of a halophyte, *Arthrocnemum indicum*, are able to colonize the root of peanut and contribute to maintaining ion homeostasis, reducing ROS production, and promoting growth under salt stress [13]. In another study, Ullah and Bano (2015) showed that *Bacillus* sp. and *Arthrobacter pascens* isolated from the rhizospheric soils of halophytes *Atriplex leucoclada* and *Suaeda fruticosa*, respectively, could increase phosphate availability, induce the accumulation of osmolytes, elevate antioxidant activity, and promote growth when inoculated into the root of maize [14]. Similarly, Xiong et al. (2019) revealed that inoculation of *Glutamicibacter halophytocola* isolated from the coastal halophyte *Limonium sinense* could increase osmolyte content, enhance antioxidant activity, and improve ion homeostasis in tomato seedlings, resulting in higher biomass and better growth under stress [15].

All of those reports suggest that bacteria associated with the roots of halophytes could be inoculated into crops to enhance resistance against salt stress and mineral deficiency. Information about halophyte microbiomes and their association with soil physical and chemical properties is pivotal for the identification of beneficial bacteria. Nevertheless, data regarding root microbiomes in halophytic plants, especially from tropical coastal areas, are still limited. Here, we studied the bacterial community in the roots of two halophytes, *Spinifex littoreus* and *Calotropis gigantea*, growing in a coastal sand dune area of Parangkusumo, Indonesia. Plants growing in the area are adapted to harsh environments, characterized by frequent sand and wind blasting, low nutrient and water availability, high temperature, lack of shade, salt spray, and high soil salinity [23,24,25]. We hypothesized that halophytes in the Parangkusumo sand dune ecosystem host symbiotic bacterial species that aid in adaptation to the arid, saline, and low-nutrient environment. To identify such bacteria, we correlated root microbiome data with soil salinity, pH, organic and nutrient content, and mineral composition. Strong correlations were observed between the root bacterial diversity and soil mineral composition, especially with soil Calcium (Ca), Titanium (Ti), Cuprum (Cu), and Zinc (Zn) content. Correlations between the abundance of various bacterial species and soil salinity and mineral content were also discovered, indicating that some bacteria may be sensitive to changes in soil salinity and mineral content. This investigation thus shed light on the bacterial communities associated with the roots of sand dune halophytes, the environmental factors that affect their composition, and potential bacterial species involved in plant tolerance to stress.

## 2. Materials and Methods

### 2.1. Study Area and Sample Collection

Root and leaf samples were collected from *S. littoreus* and *C. gigantea*, which grow wild in the sand dune area of Parangkusumo, Yogyakarta, Indonesia. Samples were collected along latitudinal gradient starting from the shoreline from six different populations where *S. littoreus* and *C. gigantea* were found living together. The populations were spread across three different sand dune areas: the coastal area located along the shoreline (population 1), the middle area (population 2 to 5), and the transitional area (population 6, the border between sand dune and farming area) (Figure 1 and Appendix A). From each population, we collected two *S. littoreus* and *C. gigantea* root samples for microbiome analysis. Soil samples were also collected from 5 to 20 cm depth and kept in clean plastic bags. All samples were immediately stored on dry ice upon collection and stored at −20 °C afterward.

### 2.2. DNA Extraction and Sequencing

Genomic DNA was extracted by homogenizing samples to powder in liquid nitrogen using a mortar and pestle. DNA was extracted from homogenized tissue according to the manufacturer’s instructions using a ZymoBIOMICS DNA Miniprep Kit (Zymo Research, Orange, CA, USA). Quality-controlled genomic DNA was used to prepare amplicon sequencing libraries. In brief, following the Illumina PCR Quantification Protocol Guide, 30 ng DNA template and 16S rRNA V3-V4 primers were used for polymerase chain reaction (PCR) (Illumina, San Diego, CA, USA). To complete library construction, all PCR products were purified with Agencourt AMPure XP beads (Beckman Coulter, Brea, CA, USA), dissolved in elution buffer, and finally labeled. Agilent 2100 Bioanalyzer was used to measure the library’s size and concentration (Agilent Technologies, Palo Alto, CA, USA). Two 300 bp paired-end runs were performed on qualified libraries on the Illumina HiSeq 2500 platform (San Diego, CA, USA).

### 2.3. Bacterial Diversity Analysis

To investigate the biodiversity of the surveyed samples, the sequencing data were analyzed using the QIIME 2 workflow [26]. After importing the raw sequencing data into QIIME 2, the raw sequencing data were demultiplexed to remove primer sequences from the reads. DADA2 was used to denoise and dereplicate the sequences [27]. Then, using the VSEARCH plugin, the clean data were clustered into groups with 99 percent similarity to the SILVA references [28,29]. Using online web tools https://bioinformatics.psb.ugent.be/webtools/Venn/ (accessed on 20 October 2021, the samples' shared operational taxonomic unit (OTU) was visualized. The sequence data were then classified using the classify-sklearn method in conjunction with SILVA taxonomy data [30]. After being classified, the sequences belonging to chloroplast and mitochondrial genomes were removed. Additionally, the Shannon diversity index was used to evaluate the samples’ alpha diversity. Principal component analysis (PCA) was performed using the ClustVis web tools http://biit.cs.ut.ee/clustvis/ (accessed on 27 October 2021) with default settings [31]. The data of the operational taxonomic unit of the surveyed samples were normalized to the total OTU number before performing PCA. The PCAs were calculated using the default method of singular value decomposition (SVD) with imputation. 

### 2.4. Soil pH, Salinity, Organic Carbon, Nitrogen, and Phosphate Measurement

As much as 10 g of soil and 50 mL of deionized water were mixed together. The mixture was then homogenized by vortexing for 30 min. The suspension was then used to measure the pH and conductivity using a pH meter electrode (Starter300, Ohaus, NJ, USA) and a conductometer (pHionLab PC10, H20 Rx, Artarmon, NSW, Australia), respectively. The total organic carbon, nitrogen, and phosphate were measured according to the ASTM D 5373-2002 standard.

### 2.5. Measurement of Soil Mineral Composition

As much as 10 g of soil was analyzed using X-Ray Fluorescence Niton^TM^ XL2 GOLDD (Thermo Scientific, Carlsbad, CA, USA) to calculate Aluminium (Al), Silicon (Si), Phosphorus (P), Potassium (K), Calcium (Ca), Titanium (Ti), Vanadium (V), Manganese (Mn), Ferrum (Fe), Cuprum (Cu), Zinc (Zn), Strontium (Sr), Zirconium (Zr), Barium (Ba), and Rhenium (Re) content in the soil. The mineral content measurements were presented as percentages of the weight of the soil.

### 2.6. Correlation Analysis between Bacterial and Soil Composition

Pearson correlation analysis was used to determine the relationship between bacterial diversity, microbial species abundance, and soil variables by using GraphPad Prism version 9.0.0 for Windows https://www.graphpad.com/ (accessed on 27 October 2021). The scatter diagrams were built on at least 6 data points for each pair of variables to observe the linearity and calculate the correlation coefficient. Pearson values (r) at or close to zero indicated no or a weak linear relationship, respectively; greater than +0.8 or less than −0.8 denoted a strong linear relationship. Factors with a strong linear relationship (|r| ≥ 0.8) and statistical significance (*p* ≤ 0.05) were selected for further evaluation.

## 3. Results

### 3.1. Taxonomic Composition of the Root-Associated Bacteria

A total of 1231 bacterial OTUs were obtained from the root of *C. gigantea*, while 1419 OTUs were obtained from *S. littoreus* (Appendix A). The assignment of bacterial OTUs revealed 25 phyla, 66 classes, 157 orders, 228 families, and 344 genera in *C. gigantea*, whereas 26 phyla, 62 classes, 137 orders, 205 families, and 332 genera were identified in *S. littoreus*. The most abundant phyla in the root of *C. gigantea* were Proteobacteria (38.25% reads), Actinobacteria (30.17% reads), Firmicutes (10.87% reads), and Bacteroidota (5.68% reads), together representing 84.97% of total reads over all samples. In the root of *S. littoreus*, the most abundant bacteria were Actinobacteria (41.13%), Proteobacteria (31.33%), Bacteroidota (7.67%), and Patescibacteria (7.25%), representing altogether 87.4% of the total reads (Appendix A). 

In *C. gigantea*, from population 2 to 6, the bacterial composition was quite similar at phylum level, where the roots were mainly colonized by Proteobacteria and Actinobacteria. A different composition was observed from population 1 that was located on the shoreline where Firmicutes was identified as the most dominant phylum. Among OTUs that could be assigned to species level, differences at species level could also be observed between population 1 and the other populations. *Bacillus idriensis* was found to be the most abundant species in population 1, while in other populations, *Actinosynnema pretiosum* and *Actinophytocola timorensis* were the two most abundant species (Figure 2 and Appendix A). 

For *S. littoreus*, in all populations, the roots were mainly colonized by Proteobacteria, Actinobacteria, Bacteroidota, and Patescibacteria. However, in population 6, which was located in the transitional zone between sand dunes and farming areas, a higher abundance of Firmicutes was observed. Looking at the species level, population 6 was also distinguishable from the other populations since it had higher bacterial diversity (Table 1 and Appendix A). Various bacterial species, including *Kibdelosporangium aridum, Pseudonocardia zijingensis, A. timorensis, Pseudonocardia eucalypti*, and *Bacillus aryabhattai*, could be identified from population 6, while in other populations the bacterial composition was mainly composed of *A. pretiosum* (Appendix A). 

Since soil chemical and physical properties differed between populations, these results suggest that bacterial composition in the roots of halophytes was strongly influenced by soil properties. Interestingly, contrasting responses were observed between bacteria associated with the roots of *C. gigantea* and *S. littoreus*. Bacteria in the root of *C. gigantea* were strongly influenced by the shoreline environment, while *S. littoreus* was strongly influenced by the transitional zone environment that was furthest away from the shoreline.

### 3.2. Population-Specific OTUs and Core Microbiome

From 1231 OTUs identified in the root of *C. gigantea*, 17 OTUs were shared between all populations. In *S. littoreus*, across 1419 identified OTUs, 40 were shared between all populations, suggesting the existence of a core microbiome in the root of *C. gigantea* and *S. littoreus*. Additionally, for both species in each population, population-specific OTUs were detected, indicating specific environmental effects on each population (Figure 3). Comparing OTUs from both plant species, 12 OTUs were found to be shared across all *C. gigantea* and *S. littoreus* populations, corresponding to genera: *Mycobacterium, Lechevalieria, Streptomyces, Bacillus, Dongia, Bosea, Devosia, Sphingomonas, Acidibacter*, and three genera from the family *Microscillaceae, Rhizobiaceae,* and *Sphingomonadaceae* (Figure 3 and Appendix A). These results infer that some taxa have been well adapted to sand dune ecosystems and can colonize the roots of different plant species that grow in different sand dune areas.

### 3.3. Diversity and Structure of Bacterial Community Associated with Halophyte Roots

For *C. gigantea*, the bacterial diversity in population 1, which was located on the shoreline, was lower compared to that of the other populations. This might be due to higher soil salinity in population 1 that negatively affected bacterial diversity (Table 1, Appendix A). The highest diversity was found in population 4, but it was comparable with that of the other populations. Similarly to *C. gigantea*, the bacterial diversity in the root of *S. littoreus* was also lowest in population 1. On the other hand, bacterial diversity in population 6 was considerably higher compared with that of the other populations. This result suggests that the transition from sandy soil to farming soil might have positive effects on bacterial diversity associated with the root of *S. littoreus* (Table 1). Results from alpha-diversity and taxonomic composition analysis were reflected in the way the population clustered in PCA. In *S. littoreus*, population 6 was visibly separated from the rest of the populations, in accordance with the higher bacterial diversity observed in this population. However, *in C. gigantea,* populations 1, 2, and 6 were separated from the others (Figure 4). In *C. gigantea*, population 1 was separated due to lower bacterial diversity (Table 1), while population 2 and 6 were separated due to the high number of unique OTUs identified in these two populations. There were 88 OTUs that were exclusively found in population 2, while in population 6, there were 184 unique OTUs (Figure 3).

### 3.4. The Effect of Soil Properties on the Bacterial Community Associated with the Roots of C. gigantea and S. littoreus

To evaluate whether soil chemical and physical properties affect the composition and abundance of the bacterial community associated with the roots of sand dune halophytes, we measured various soil properties across the six sampled populations. Detailed values of the measured soil properties are shown in Appendix A. Pearson correlation analysis was used to determine correlations between soil properties and alpha diversity. Although lower bacterial diversity was observed in population 1, which had higher soil salinity, in both plant species we did not find a statistically significant correlation between alpha diversity and soil salinity (*p* < 0.05). We also did not find a significant correlation between alpha diversity and soil pH, organic carbon, nitrogen, or phosphorus content (Table 2). These results suggest that none of those factors had a significant impact on overall diversity. Nevertheless, we found strong correlations between soil mineral composition and bacterial diversity. In *C. gigantea*, a strong positive correlation was observed between alpha diversity and soil Zn level (r = 0.914, *p* < 0.05). Similarly, in *S. littoreus,* a positive correlation was also observed between bacterial diversity and soil Zn (r = 0.881) and Ti (r = 0.829) content. On the other hand, soil Ca (r = −0.907) and Cu (r = −0.823) levels seemed to have a negative effect on the diversity of bacteria associated with the root of *S. littoreus* (Table 2). These results showed that soil mineral composition, especially Zn, Ti, Ca, and Cu content, was the main factor influencing bacterial diversity associated with the roots of sand dune halophytes. 

Despite having no effect on overall diversity, soil salinity might have significant effects on the abundance of specific bacteria. To test this hypothesis, we performed a Pearson correlation analysis between soil properties and species relative abundance. In *C. gigantea*, we observed a strong positive correlation between the abundance of *B. idriensis* and soil salinity (r = 0.961) and phosphorus (r = 0.915) content, suggesting that this bacterium can thrive in saline soil with high phosphorus concentrations (Table 3). In *S. littoreus*, the only species that correlated with salinity was *Pseudolabrys taiwanensis,* which showed a negative correlation (r = −0.871) (Table 4).

Soil mineral content, especially of soil Ca, Cu, Ti, and Zn, seems to have a strong effect on the abundance of several bacterial species associated with the root of *S. littoreus*. Soil Ca and Cu had a negative correlation with the abundance of *P. eucalypti, A. timorensis, B. aryabhattai,* and *P. zijingensis*. On the other hand, the four bacteria showed strong positive correlations with soil Ti and Zn. Interestingly, *A. pretiosum*, the most abundant bacterium found in the root of *S. littoreus*, showed a completely opposite trend; it was positively correlated with Ca and Cu but negatively correlated with Ti and Zn (Table 4). Altogether, these results suggest a highly unique and specific effect of each mineral on bacterial abundance.

## 4. Discussion

Plants growing in the coastal sand dune area of Parangkusumo, Indonesia, are frequently exposed to high soil salinity, high temperature, low nutrients, and limited water availability [23,24,25]. The harsh soil and environmental conditions in the area provide interesting models to study the role of root microbiota in plant adaptation to high soil salinity and nutrient deficiency. Using models *C. gigantea* and *S. littoreus* populations in their natural habitat, we studied the dynamics of the root microbiota along a latitudinal gradient and changes in soil physicochemical properties.

We found that the overall most abundant phyla in both plants were Proteobacteria and Actinobacteria; this result was in accordance with previous studies of rhizospheric microbiota in coastal [32] and desert soils [33], suggesting that members of these phyla can thrive in arid environments. In both plants, the most abundant identified species was *A. pretiosum*; this bacterium was first isolated from leaf surface of *Carex* sp. and is known to produce ansamitocin, a potent antibiotic and antitumor compound [34,35]. *A. pretiosum* is also reported to be associated with the root of *Putterlickia verrucosa*, a shrub that is distributed in the coastal areas of South Africa, Eswatini and Mozambique [36]. Most of the studies regarding *A. pretiosum* have been focused on its culture strategies, metabolic pathways, and ansamitocin producing ability [37,38,39,40,41,42]; however, there is still no information regarding the role of *A. pretiosum* in plant growth and response to environmental stresses. Since *A. pretiosum* was abundantly found in the root of halophytic *C. gigantea* and *S. littoreus*, it would be interesting to evaluate whether its association with halophyte roots contributes to plant tolerance against salinity and nutrient deficiencies. Beside *A. pretiosum*, several bacteria with known plant growth promoting activity, such as *K. aridum, B. idriensis*, and *B. aryabhattai*, were also detected in high abundance in the roots of *C. gigantea* and *S. littoreus*. These bacteria can produce 1-aminocyclopropane-1-carboxylic acid deaminase (ACCD) and a range of phytohormones that have a positive effect on plant growth [43,44]. Nevertheless, their biological role in plant tolerance against salinity and nutrient deficiency has never been investigated before.

Our data showed that soil physicochemical properties along the coastal sand dune latitudinal gradient clearly shaped the composition and diversity of the root-associated microbial community. In both plants, soil Zn content was positively correlated with root microbial diversity. In accordance, Pan et al. (2020) also reported a positive association between soil Zn level and soil bacterial diversity. One hypothesis is that a high level of Zn in soil might induce the emergence of zinc-tolerant bacteria, resulting in higher diversity [45,46]. Besides Zn, bacterial diversity in the root of *S. littoreus* was also affected by soil titanium (Ti), copper (Cu), and calcium (Ca) content. Like Zn, we also found that Ti had a positive effect on root bacterial diversity. In contrast to our findings, previous studies reported that the application of Ti into soil did not affect soil bacterial diversity in pitaya [47] and wheat [48] fields, while in grape field, application of Ti was reported to decrease bacterial diversity [47]. Ti can either stimulate or inhibit plant growth, depending on the plant species [49]. Since Ti's effect on plant physiology occurred in a species-specific manner, the effect of Ti on soil and root microbial diversity may also depend on the host plant species. In contrast to Zn and Ti, soil Cu and Ca levels were negatively associated with root microbial diversity. The negative effect of Cu on microbial diversity is well known. Excess levels of Cu are toxic to bacteria because membrane-bound Cu can catalyze the formation of free radicals [50,51]. Calcium can have positive or negative effects on microbial diversity depending on the environmental setting. In acidic soil, increasing Ca could have a positive effect on microbial diversity [52,53], but in saline or karst environments, Ca could negatively influence microbial diversity due to excess salinity and soil pH [54].

Although no statistically significant correlation was observed between soil salinity and root microbial diversity, significant associations were found between the abundance of several bacterial species and salinity. In *C. gigantea*, a strong positive correlation was observed between *B. idriensis* and soil salinity, suggesting that this bacterium has salt-tolerant properties. Afzal et al. (2017) showed that *B. idriensis* isolated from the wild shrub *Dodonaea viscosa* L. could promote root growth when inoculated into canola [21], but its role in enhancing tolerance to salt has never been investigated before. Besides affecting microbial diversity, the abundance of several bacterial species was also strongly influenced by soil Zn, Ti. Cu, and Zn content. We observed the antagonistic effect of these four minerals on different bacterial species. Ca and Cu exhibited a negative influence on the abundance of *P. eucalypti, A. timorensis, B. aryabhattai,* and *P. zijingensis*, while Zn and Ti showed an opposing effect. Among these bacteria, *B. aryabhattai* has been reported to promote plant growth and tolerance against heat stress [55]. Interestingly, the effect of Zn, Ti, Cu, and Zn was completely reversed for *A. pretiosum*; it had a positive correlation with Ca and Cu and a negative correlation with Ti and Zn. It is well known that soil minerals can affect bacterial growth in a taxon-specific manner. These mineral-associated bacteria might influence the biogeochemical cycling of the associated minerals and nutrient availability for the host plants [56,57,58]. Further work is required to evaluate whether *P. eucalypti, A. timorensis, B. aryabhattai, P. zijingensis*, and *A. pretiosum* could together regulate Zn, Ti, Cu, and Zn availability in coastal sand dune soil and affect the growth of host plants. 

Note that in this study, potential bacteria were listed from correlation analysis between bacterial abundance and soil salinity level or mineral content. Further work is required to validate the biological role of these root-associated bacteria in plant tolerance against stresses. Previous studies have shown that bacteria isolated from the roots of halophytes could enhance plant tolerance against salt stress when inoculated into crops, possibly by helping plants maintain osmotic balance and reduce cellular oxidative stress [13,14,15]. Due to its strong positive correlation with soil salinity level, we expect *B. idriensis* to serve similar functions. Thus, future work will be focused on the isolation, culture, and transplantation of potential bacteria identified in this study, to confirm that colonization of roots by these bacteria could provide plants with improved tolerance against salinity and nutrient deficiency. Overall, the information gathered from this study can be used as a basis for further validation experiments and biotechnological applications, especially for amelioration of abiotic stresses in plants using beneficial microbes.

## 5. Conclusions

Soil salinization and mineral deficiency are major abiotic stresses that can decrease plant productivity. Several studies have shown that root-associated microorganisms can enhance tolerance to salt stress and increase nutrient availability through multiple mechanisms. Our exploratory work showed that the bacterial community in the roots of coastal sand dune halophytes *S. littoreus* and *C. gigantea* was strongly influenced by soil physicochemical properties, especially by soil salinity and Zn, Ti, Cu, and Ca content. Based on correlation analysis between bacterial abundance and soil properties, *B. idriensis* was identified as a potential salt-tolerant bacterium that might be involved in plant tolerance against salt stress, while *P. eucalypti, A. timorensis, B. aryabhattai, P. zijingensis*, and *A. pretiosum* were identified as taxa that were responsive to soil Zn, Ti, Cu, and Ca content. Future work will focus on the isolation, culture, and transplantation of these potential bacteria to validate their involvement in nutrient cycling and resistance against salinity.

## Figures and Tables

**Figure 1 biology-11-00695-f001:**
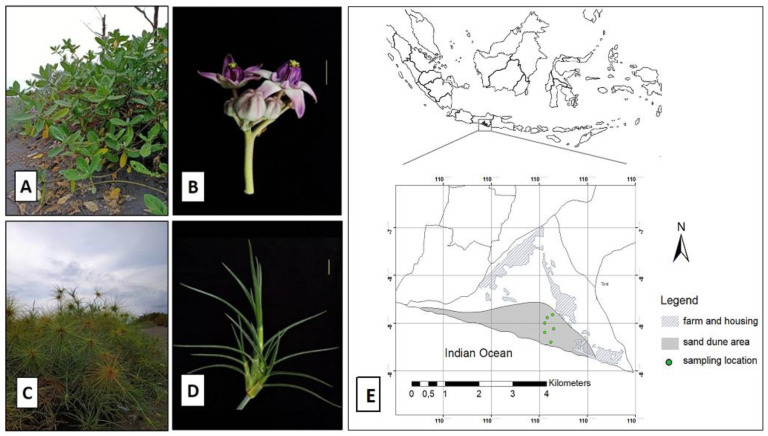
Halophyte habitats and sampling locations. (**A**) *Calotropis gigantea* at Parangkusumo sand dune and (**B**) its flowering part. (**C**) *Spinifex littoreus* at Parangkusumo sand dune and (**D**) its leaf morphology. (**E**) Six sampling locations across Parangkusumo sand dune area.

**Figure 2 biology-11-00695-f002:**
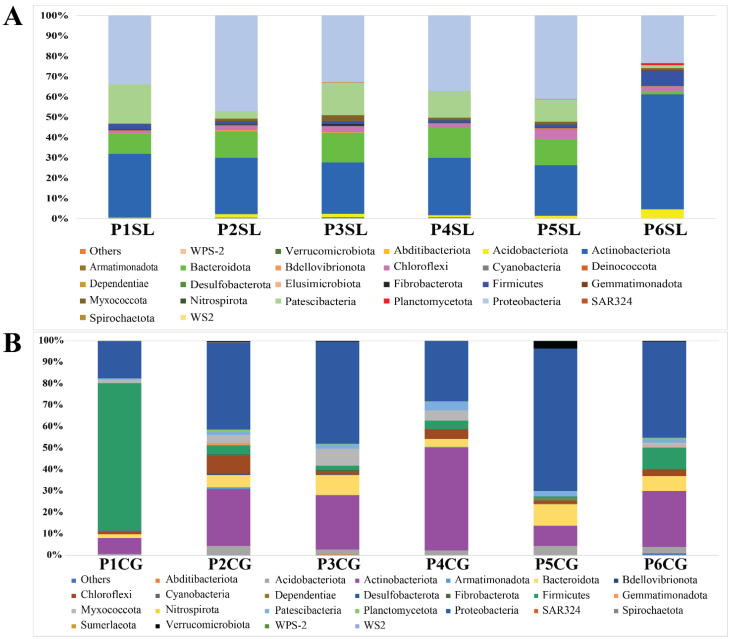
Bacterial taxonomic composition and relative abundance at phylum level. (**A**) Relative abundance of bacteria associated with the root of *S. littoreus* across six different sampling populations at phylum level. (**B**) Relative abundance of bacteria associated with the root of *C. gigantea* across six different sampling populations at phylum level. Abbreviations of sampled populations: P—Population (Population 1 to 6, P1 to P6), CG—*C. gigantea*, SL—*S. littoreus*.

**Figure 3 biology-11-00695-f003:**
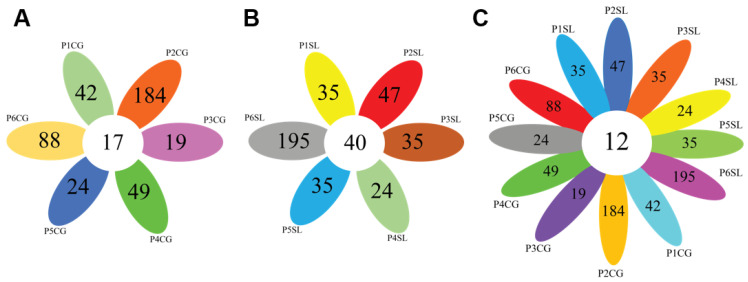
Venn diagrams comparing the number of population-specific and population-shared OTUs. (**A**) Comparison across different populations of *Calotropis gigantea*, (**B**) Comparison across different populations of *Spinifex littoreus*, (**C**) Comparison across all *C. gigantea* and *S. littoreus* populations. Abbreviations of sampled populations: P-Population (Population 1 to 6, P1 to P6), CG—*C. gigantea*, SL—*S. littoreus*.

**Figure 4 biology-11-00695-f004:**
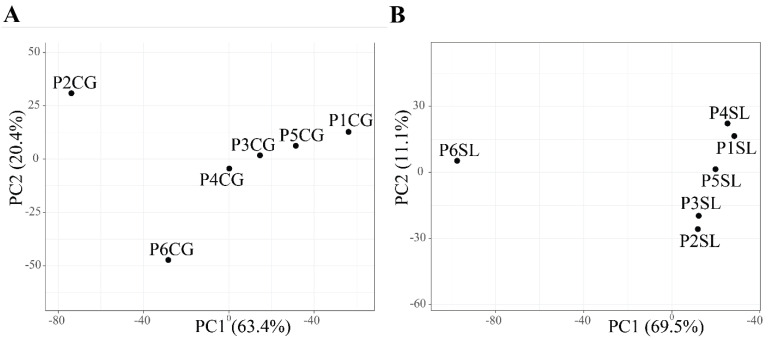
Principal component analysis of obtained bacterial OTUs present in the roots of (**A**) *C. gigantea* and (**B**) *S. littoreus*. Abbreviations of sampled populations: P—Population (Population 1 to 6, P1 to P6), CG—*C. gigantea*, SL—*S. littoreus*.

**Table 1 biology-11-00695-t001:** Species alpha diversity in sampling populations as measured by Shannon Index.

Species	Population	Shannon Index
*C. gigantea*	P1CG	3.39
	P2CG	5.11
	P3CG	5.27
	P4CG	5.91
	P5CG	4.14
	P6CG	5.82
*S. littoreus*	P1SL	5.34
	P2SL	6.07
	P3SL	5.99
	P4SL	5.90
	P5SL	5.91
	P6SL	7.34

**Table 2 biology-11-00695-t002:** Correlation Analysis Between Bacterial Diversity and Soil Properties.

Soil Properties	Correlation *C. gigantea* (R)	Correlation *S. littoreus* (R)
pH	−0.305	−0.588
Salinity	−0.704	−0.613
Organic carbon	0.394	0.118
Nitrogen	−0.0379	0.532
Phosphorus	−0.682	−0.665
Calcium (Ca)	−0.792	−0.907 *
Titanium (Ti)	0.759	0.829 *
Cuprum (Cu)	−0.467	−0.823 *
Zinc (Zn)	0.914 *	0.881 *

Asterisk represent significant correlation at *p* < 0.05 (Pearson correlation coefficient).

**Table 3 biology-11-00695-t003:** Correlation between Bacterial Abundance Associated with the Root of *C. gigantea* and Soil Properties.

Species	Salinity (R)	Carbon (R)	Phosphorus (R)	Calcium (R)	Titanium (R)	Cuprum (R)	Zinc (R)
*A._timorensis*	−0.201	−0.103	−0.082	−0.161	0.269	0.020	0.156
*Actinosynnema_pretiosum*	−0.273	0.206	−0.393	−0.157	0.143	0.009	0.146
*B._idriensis*	0.962 *	−0.521	0.915 *	0.571	−0.420	0.433	−0.685

Asterisk represent significant correlation at *p* < 0.05 (Pearson correlation coefficient).

**Table 4 biology-11-00695-t004:** Correlation Between Bacterial Abundance Associated with the Root of *S. littoreus* and Soil Properties.

Species	Salinity (R)	Carbon (R)	Phosphorus (R)	Ca (R)	Ti (R)	Cu (R)	Zn (R)
*Rhizobium lusitanum*	−0.062	0.873 *	0.046	0.160	−0.368	0.388	−0.432
*Pseudonocardia eucalypti*	−0.310	−0.181	−0.387	−0.868 *	0.815 *	−0.943 *	0.845 *
*Pseudolabrys taiwanensis*	−0.871 *	0.182	−0.791	−0.673	0.659	−0.135	0.637
*Mitsuaria chitosanitabida*	0.016	0.898 *	0.098	0.202	−0.408	0.404	−0.490
*Dongia mobilis*	−0.776	0.097	−0.850 *	−0.560	0.566	−0.170	0.590
*Actinophytocola timorensis*	−0.321	−0.175	−0.396	−0.878 *	0.827 *	−0.937 *	0.851 *
*Bacillus aryabhattai*	−0.325	−0.183	−0.402	−0.866 *	0.809 *	−0.947 *	0.851 *
*Pseudonocardia zijingensis*	−0.311	−0.181	−0.388	−0.870 *	0.817 *	−0.942 *	0.847 *
*TM7 phylum*	0.586	−0.448	0.462	0.551	−0.416	0.185	−0.394
*Amycolatopsis australiensis*	−0.724	−0.374	−0.705	−0.449	0.421	−0.450	0.709
*Ralstonia mannitolilytica*	−0.346	−0.049	−0.267	0.321	−0.439	0.160	−0.079
*Kibdelosporangium aridum*	−0.384	0.205	−0.418	−0.811 *	0.665	−0.771	0.679
*Actinosynnema pretiosum*	0.459	0.102	0.551	0.837 *	−0.723 *	0.943 *	−0.857 *

Asterisk represent significant correlation at *p* ≤ 0.05 (Pearson correlation coefficient).

## Data Availability

The sequencing data presented in this study are openly available in National Center for Biotechnology Information (NCBI), BioProject number PRJNA813219.

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
