# Peer review of "Soil Mineral Composition and Salinity Are the Main Factors Regulating the Bacterial Community Associated with the Roots of Coastal Sand Dune Halophytes"

_biology, 2022, doi:10.3390/biology11050695_

Round 1

Reviewer 1 Report

Reviewer report for research article entitled "Soil Mineral Composition and Salinity are the Main Factors Regulating Bacterial Community Associated with the Root of Coastal Sand Dune Halophytes'. The article is focused on the microbial diversity of endophytes of two plants which grow under high soil salinity. Authors also try to correlate between soil mineral composition and microbial root diversity. The study was focused on two plants Spinifex littoreus  and Calotropis gigantea grown on different area of coasatal sand dune, Indonesia. 

This type of research is of high interest for both scientific and plantation society. However, this research was based on sampling following directly by bacterial diversity analysis based on DNA extraction and sequencing data. Therefore, these data can not be validated to ensure that these claimed strains are responsible for the development of resistance of plant.

Major Comment:

For this type of research, next step is a MUST to isolate the proposed endophyte strains which claimed to support halotolerant and re-inoculated in the plant to ensure that the halotolerance is really due to the presence of these strains in the root. In addition, experiment need to be carried out of inoculated roots grown under high salinity and following the bacterial diversity analysis to ensure that these strains are responsible for salt stress or mineral stress tolerance. Otherwise, this research is considered as incomplete and the results/conclusions are not validated. 

Minor comments: 

  • For figure 1E, its necessary to provide some information about the selection of sampling area and how to claim it represent this zone. In addition, the map of Indonesia need to be provided and put this map as magnification part.
  • For all instrument used need to provide (Model, Company, City, Country).
  • The discussion need to be further improved 
  • To conclude that B. idriensis as potential salt-tolerant bacteria and involve in plant tolerance against salt stress as root endophyte, and some other taxa are responsive to soil Zn, Ti, Cu, and Ca content (as mentioned), its necessary to do plantation experiments using those microbes as inoculants before coming to this conclusion. 

Author Response

The article is focused on the microbial diversity of endophytes of two plants which grow under high soil salinity. Authors also try to correlate between soil mineral composition and microbial root diversity. The study was focused on two plants Spinifex littoreus and Calotropis gigantea grown on different area of coasatal sand dune, Indonesia. 

This type of research is of high interest for both scientific and plantation society. However, this research was based on sampling following directly by bacterial diversity analysis based on DNA extraction and sequencing data. Therefore, these data can not be validated to ensure that these claimed strains are responsible for the development of resistance of plant.

Major Comment:

For this type of research, next step is a MUST to isolate the proposed endophyte strains which claimed to support halotolerant and re-inoculated in the plant to ensure that the halotolerance is really due to the presence of these strains in the root. In addition, experiment need to be carried out of inoculated roots grown under high salinity and following the bacterial diversity analysis to ensure that these strains are responsible for salt stress or mineral stress tolerance. Otherwise, this research is considered as incomplete and the results/conclusions are not validated. 

Response:

We are pleased to know that the first reviewer thought the paper would be of high interest for both scientific and plantation society. This work is intended as exploratory work to characterize the bacterial community associated with the root of sand dune halophytes and the soil physicochemical properties that affecting their composition. Based on strong statistical correlation between soil properties and bacterial abundance we suggest several bacterial species that might involve in tolerance against salinity and nutrient deficiency, however we are not making claim in the manuscript that these bacteria are serving those biological function in the root of Spinifex littoreus and Calotropis gigantea. We rewrote our summary (line 31-34), abstract (line 64-66), discussion, and conclusion to make this clearer.

We agree with the reviewer that the next follow up experiment is to isolate the potential bacteria, developing culture methods for the bacteria, and inoculating it into halophytes and crops that are subjected to stresses. These experiments will confirm the role of endophytic bacteria in tolerance against stresses. Currently we are doing these experiments but it will take months to finish the experiments due to technical challenges and the scale of the experiment. For that reason, we cannot include the requested data in this manuscript.

We found that it is common to separate the exploratory part of the project and biological validation of the microbial function in different publication. As we aware, this is the first report for microbial communities associated with the root of Spinifex littoreus and Calotropis gigantea, two halophytes that are commonly found and widely distributed in tropical coastal area. This is also the only few reports describing bacterial communities associated with the unique tropical sand dune ecosystem and their association with various soil physicochemical properties. Therefore, we believe that our work has enough novelties and information to be published at Biology.

Minor comments: 

1. For figure 1E, its necessary to provide some information about the selection of sampling area and how to claim it represent this zone. In addition, the map of Indonesia need to be provided and put this map as magnification part.

Response: Thank you for the suggestion, we provided additional information in the method regarding the sampling area (line 139-142), we also revise Figure 1E to include the map of Indonesia and the sampling location.

2. For all instrument used need to provide (Model, Company, City, Country).

Response: Thank you for the correction, we revised the manuscript accordingly to include those information

3. The discussion need to be further improved

Response: We extend our discussion to discuss the limitation of the work and the follow-up experiments (line 436-445)

4. To conclude that B. idriensis as potential salt-tolerant bacteria and involve in plant tolerance against salt stress as root endophyte, and some other taxa are responsive to soil Zn, Ti, Cu, and Ca content (as mentioned), its necessary to do plantation experiments using those microbes as inoculants before coming to this conclusion.

Response: We rewrote our conclusion to make it clearer that this is an exploratory work and further work is required to validate the biological function of potential bacteria (line 451-460).

Reviewer 2 Report

dear authors, thanks for your manuscript. I provide little suggestions:

summary: I think that you should rewrite it, especially the first line

methods: rewrite and expand the section for statistic. in the results you reported pca analysis but you did not describe it in the methods

Author Response

Dear authors, thanks for your manuscript. I provide little suggestions:

1. Summary: I think that you should rewrite it, especially the first line

Response: Thank you for the positive feedback and the suggestions, we rewrote the summary to make it less technical and more understandable for general public.

2. Methods: rewrite and expand the section for statistic. in the results you reported pca analysis but you did not describe it in the methods

Response: Thank you for the suggestion, we revised the method section, especially in the Bacterial Diversity Analysis and Correlation Analysis sub-section to better describe the statistical analysis we applied.

Reviewer 3 Report

Good paper, valuable to be published in this Journal. Small technical comments I added in the paper.

Author Response

Good paper, valuable to be published in this Journal. Small technical comments I added in the paper

Response: Thank you for your encouraging words, we are pleased to know that reviewer think the manuscript is valuable and suitable to be published at Biology. Thank you for the technical comments, we revised our manuscript according to your suggested corrections (please see the revised version of our manuscript).

Round 2

Reviewer 1 Report

In the second revision, authors did good effort to address or minor issues raised in my first round of revision. However, the main issue raised was still not addressed which is related to validation to confirm that the identified microbes are related to salt tolerant of the plant which need to isolate these stain, propagate and incoulate to plant to ensure that these microbes are related to plant halotolerant and other stresses. 

I think in the reply of authors, they also agree and they are currently doing these experiments (as in their reply). Without these experiments we can not validate the link of the presence of these microbes and halotolerant. Therefore, I consider this work is complete and can not provide evidence of the claimed relationship between plant resistance and the presence of these microbes. 

Thus, I do not recommend to publish this work without adding the experimental work of plant inoculation with these microbes.